# Effects of Supplementing the Usual Diet with a Daily Dose of Walnuts for Two Years on Metabolic Syndrome and Its Components in an Elderly Cohort

**DOI:** 10.3390/nu12020451

**Published:** 2020-02-11

**Authors:** Ahmed Al Abdrabalnabi, Sujatha Rajaram, Edward Bitok, Keiji Oda, W. Lawrence Beeson, Amandeep Kaur, Montserrat Cofán, Mercè Serra-Mir, Irene Roth, Emilio Ros, Joan Sabaté

**Affiliations:** 1Center for Nutrition, Healthy Lifestyle and Disease Prevention, School of Public Health, Loma Linda University, Loma Linda, CA 92350, USA; aalabdrabalnabi@llu.edu (A.A.A.); srajaram@llu.edu (S.R.); koda@llu.edu (K.O.); lbeeson@llu.edu (W.L.B.); akaur1@llu.edu (A.K.); 2Department of Nutrition and Dietetics, School of Allied Health Professions, Loma Linda University, Loma Linda, CA 92350, USA; ebitok@llu.edu; 3Lipid Clinic, Endocrinology and Nutrition Service, Hospital Clínic, Institut d’Investigacions Biomèdiques August Pi i Sunyer (IDIBAPS), 08036 Barcelona, Spain; MCOFAN@clinic.cat (M.C.); SERRAMIR@clinic.cat (M.S.-M.); ROTH@clinic.cat (I.R.); EROS@clinic.cat (E.R.); 4CIBER Fisiopatología de la Obesidad y Nutrición (CIBEROBN), Instituto de Salud Carlos III, 28029 Madrid, Spain

**Keywords:** metabolic syndrome, blood pressure, fasting blood glucose, triglycerides, walnuts

## Abstract

Accumulating evidence links nut consumption with an improved risk of metabolic syndrome (MetS); however, long-term trials are lacking. We examined the effects of a daily dose of walnuts for two years on MetS in a large elderly cohort. A total of 698 healthy elderly participants were randomly assigned to either a walnut supplemented or a control diet. The participants in the walnut group were provided with packaged walnuts (1, 1.5, or 2 oz. or ~15% of energy) and asked to incorporate them into their daily habitual diet. The participants in the control group were asked to continue with their habitual diet and abstain from eating walnuts and other tree nuts. Intake of n-3 fatty acid supplements was not permitted in either group. Fasting blood chemistries, blood pressure, and anthropometric measurements were obtained at baseline and at the end of intervention. A total of 625 participants (67% women, mean age 69.1 y) completed this two-year study (90% retention rate). Triglycerides decreased in both walnut (−0.94 mg/dl) and control (−0.96 mg/dl) groups, with no significant between-group differences. There was a non-significant decrease in systolic and diastolic blood pressure in the walnut group (−1.30 and −0.71 mm Hg, respectively) and no change in the control group. Fasting blood glucose decreased by ~1 point in both the walnut and control groups. There were no significant between-group differences in the development or reversion of MetS. In conclusion, supplementing the diet of older adults with a daily dose of walnuts had no effect on MetS status or any of its components, although the walnut group tended to have lower blood pressure.

## 1. Introduction

Metabolic syndrome (MetS) is a cluster of risk factors that increases an individual’s risk of developing cardiovascular disease (CVD) and type-2 diabetes mellitus (T2DM) [1]. The coexistence of at least three of the following: abdominal obesity, high triglycerides, low high-density lipoprotein cholesterol (HDL-C), high blood pressure, and/or high fasting blood sugar characterizes MetS [2,3].

The worldwide prevalence of MetS ranges from 20 to 30% of the adult population [4]. In the United States (U.S.), the prevalence is higher, affecting 34.2% of adults [5]. Although MetS prevalence has remained stable over the last decade, the proportion of people with abdominal obesity and high blood glucose levels has continued to rise [6,7]. This rise is likely to lead to an increase in the prevalence of chronic diseases, such as T2DM and CVD [7], and a subsequent increase in healthcare costs. Preventing and treating MetS through lifestyle modification is an integral strategy for reducing its burden and that of associated comorbidities [8].

Diet and exercise have been proposed as the main strategies for the prevention of MetS [8]. Concerning diet, vegetarian [9] and Mediterranean diets [10,11] have been associated with decreased risk of developing MetS. A common characteristic of these dietary patterns is the customary intake of tree nuts (henceforth referred to as nuts). The most commonly consumed nuts include walnuts, pistachios, cashews, almonds, and hazelnuts [12,13]. The results from a meta-analysis and systematic review of 61 controlled intervention trials showed that a higher consumption of tree nuts lowered selected CVD risk factors, including total cholesterol, low-density lipoprotein cholesterol (LDL-C), and triglycerides [14]. The results from another meta-analysis of 49 randomized clinical trials showed that nut consumption modestly decreased blood glucose and triglycerides [15]. The beneficial effects of nuts are attributed to their unique nutrient profile, which includes fiber, vegetable protein, monounsaturated fat (MUFA), polyunsaturated fat (PUFA), vitamin E, magnesium, and other bioactive components [12,13]. Walnuts in particular are a rich source of PUFA [α-linolenic acid (ALA) and linoleic acid (LA)], as well as bioactive phytonutrients, such as polyphenols [16,17,18]. Few studies indicate that regular consumption of walnuts alone [19,20] or mixed nuts with a predominance of walnuts in the context of a Mediterranean diet [21] may confer beneficial effects on MetS and its components. To date, the effects of regular walnut consumption on MetS have not been assessed in healthy elderly individuals.

We had a unique opportunity to examine the effects of walnuts on MetS over a two-year trial in a free-living healthy elderly population participating in the Walnuts and Healthy Aging (WAHA) study. We were primarily interested in determining the incidence and reversion rates of MetS in addition to assessing the effect of walnuts on individual MetS components.

## 2. Materials and Methods

### 2.1. Study Design and Intervention

This study is a secondary data analysis within the WAHA study, a two-year, dual center, randomized clinical trial examining the effects of daily walnut consumption on age-related cognitive decline and macular degeneration [22]. In brief, a total of 708 healthy elderly individuals aged 62 to 79 years were recruited from Loma Linda, California (LLU) and Barcelona, Spain (BCN). The participants were eligible for the study if they were reasonably healthy, able to read and write in English or Spanish (depending on location), and able to attend bimonthly follow-up clinic visits. The WAHA study was conducted in accordance with the guidelines of the Declaration of Helsinki and the Ethics Committee of each center approved it. All participants provided written informed consent before enrollment. Those who were extremely obese [body mass index (BMI) ≥ 40 kg/m^2^], had uncontrolled diabetes or hypertension, or were allergic to walnuts were excluded from the study. 

Prior to randomization, subjects who met eligibility criteria completed a three-day food record and filled in a physical activity questionnaire. Subjects were then randomized to either the walnut or control group. Those in the walnut group were provided with packaged walnuts equivalent to approximately 15% of their daily energy needs, to incorporate into their daily habitual diet. Walnut dosage was determined utilizing the World Health Organization energy needs equation for adults >60 y [23]. Based on this calculation, individuals received 30, 45, or 60 g (1, 1.5, or 2 servings, respectively) of walnuts per day. The walnuts were provided as eight-week allotments at each of their bi-monthly clinic visits with the research dietitian. The participants were not given instructions on how to consume walnuts or strategies for replacing other foods with them. Those in the control group simply continued with their habitual diet, but were instructed to abstain from eating walnuts or consuming more than two servings of other types of nuts per week. Both groups were advised to avoid n-3 fatty acid supplements, reduce EPA/DHA sources, and reduce other ALA sources, such as flaxseed and canola oil. This study includes data from the two study sites (LLU & BCN).

### 2.2. Measurements

Body mass index (BMI) was calculated from height and weight, which were collected at baseline and at end of study. Height was measured with a wall-mounted stadiometer. Body weight was determined with a digital precision scale. Blood pressure was measured at baseline and at scheduled bimonthly clinic visits using an Omron BP762 Series blood pressure cuff. A total of three blood pressure measurements were obtained at three-minute intervals and the mean of the last two measurements was recorded [24,25]. Overnight fasting blood samples were collected at baseline and at year 2 and kept frozen at –80 °C until analysis. Upon study completion, all of the samples were shipped to the appropriate laboratory for biochemical determinations. All samples were concurrently run in the same laboratory to reduce assay variability. Lab technicians were blinded to the intervention group.

### 2.3. Determination of MetS

MetS was defined according to the harmonized criteria of the Adult Treatment Panel III (ATP III), which requires the presence of at least three of the following five components for diagnosis: (a) abdominal obesity (waist circumference) larger than 102 cm for men and 88 cm for women; (b) triglyceride levels of 150 mg/dL or higher, or taking fibrate medication; (c) HDL-C of less than 40 mg/dL for men and 50 mg/dL for women; (d) blood pressure of 130/85 mmHg or higher, or taking antihypertensive drugs; and, (e) fasting blood sugar of 100 mg/dL or higher, or taking antidiabetic medications [2,3]. In this study, we utilized BMI as a surrogate for waist circumference due to the large number of missing waist circumference data points. A study by Hivert et al. reported that a BMI of ≥ 29.1 kg/m^2^ in men and ≥ 27.2 kg/m^2^ in women could be used as surrogate measures to define central obesity in the absence of waist circumference data [24].

### 2.4. Compliance and Retention

The participants visited the research center bi-monthly throughout the two-year study period, during which they were provided with a two-month supply of walnuts, depending on group assignment. Further, the research dietitian assessed participants for subjective compliance to study protocol, potential side effects, and offered motivation to encourage retention. Those in the walnut group were provided with extra walnuts to take family needs into consideration. Quarterly educational group activities were offered to participants in the control group to improve retention, since no active intervention was provided to them. Objective compliance to the intervention was determined by using nutrient analysis from dietary recalls and by assessing the red blood cell content of ALA, a fatty acid characteristic of walnuts [18], as proportion of total identifiable fatty acids [26]. Dietary data were used to verify whether the participants in the walnut group consumed their daily allotted amount of walnuts and whether those in the control group abstained from eating walnuts altogether. 

### 2.5. Measuring Tools

Dietary intake was assessed while using five unannounced 24-hour diet recalls (LLU) or three-day food records that were obtained every six months (BCN) over the two years of the study. Participants answered an abbreviated version of the Minnesota Leisure Time Physical Activity Questionnaire to evaluate their level of physical activity, which has been previously validated in the United States [27] and Spain [28]. The outcomes that we assessed in the present study were secondary to the original WAHA study. Therefore, we could not decide which tools to utilize in assessing dietary intake and physical activity level. Subsequently, we computed metabolic equivalent (MET) per week using the Centers for Disease Control and Prevention (CDC) and American College of Sports Medicine guidelines [29].

### 2.6. Statistical Methods and Analyses

The subjects in both groups were classified as either having or being free of MetS at baseline and at two years. The MetS incidence rate was computed among those who were free of MetS at baseline and developed MetS during the two years of follow-up. Additionally, the reversion rate was computed after two years among those who were classified as having MetS at baseline but were free of MetS after follow-up.

Marginal logistic models using generalized estimating equations (GEE) were used to compare the effect of supplementing the diet with walnuts for two years (compared to control) on the prevalence of MetS. Logistic regression was used to evaluate MetS development or reversion after 2 years among the walnut treatment and control groups. Center, sex, age, weight, physical activity, smoking status, and hypolipidemic, antihypertensive, and antidiabetic drug use were added to the models to account for potential confounding by these factors.

Chi-square contingency tests were used to compare the categorical and qualitative variable distributions between the intervention and control groups. Mixed linear models were used to investigate changes in blood pressure, BMI, HDL-C, triglycerides, and fasting blood glucose, accounting for both within person (time effect) and between treatment effects adjusting for baseline status of MetS and relevant demographics. Statistical significance was achieved at *P* < 0.05. All analyses were performed using Statistical Analysis System (SAS Version 9.4, SAS Institute, Cary, NC).

When applying *t*-tests to compare means across groups, skewness and kurtosis measurements were assessed in tests for normality. For mixed linear models on changes in MetS components, the assumption of normality was assessed by a visual inspection of residual plots and normal probability plots. A log-transformation was applied when residuals were right-skewed. For such outcomes, differences of adjusted means on the log scale were back-transformed, yielding mean ratios. 

## 3. Results

### 3.1. Demographic Profile of Participants

Table 1 shows the baseline characteristics of 625 participants for whom data were available (LLU, *n* = 299 and BCN, *n* =326); mean age = 69.1; 67% female. While the retention rate of the study was 90%, seventy-three participants (40% from walnut group) dropped out due to loss to follow-up, voluntarily left study, health issues (i.e., cancer, kidney-related events, stroke), wish to eat walnuts, digestive intolerance to walnuts, and death (*n* = 1) that were unrelated to the study interventions. The dropouts were not significantly different from completers. As for completers, there were no significant differences between the walnut and control groups regarding baseline characteristics. 

### 3.2. Nutrient Profile

Table 2 displays the mean daily intake of energy and selected nutrients in the walnut and control groups at two years. Overall, the participants in the walnut group had significantly greater intake of total energy and all macronutrients except for total carbohydrate. As expected from the composition of walnuts, intake of dietary PUFA (i.e., ALA and LA) was significantly higher in the walnut group when compared to the control group. Figure 1 shows the classification of the development and regression rates of MetS. Evaluating the nutritional quality of study diets and examining nutrient displacement related with walnut enrichment was reported in our previous study [30]. 

### 3.3. Changes in Metabolic Syndrome Components

Table 3 shows data on the effects of walnut consumption on each component of MetS. When compared to controls, there was no significant effect of supplementing the diet with walnuts on individual components of MetS. Even so, a significant decrease in plasma triglycerides and HDL-C was observed in both groups (*P* < 0.05). The walnut group showed a greater decrease in systolic (Δ mean 2 y minus baseline = −1.30, *P* = 0.27) and diastolic (Δ mean 2 y minus baseline = −0.71, *P* = 0.37) blood pressure as compared with the control group. We evaluated the incident and reversion rate of each component of MetS, but we did not find any significant difference in any of the parameters. Therefore, we only report the continuous variables that are presented in Table 3.

### 3.4. Changes in MetS Status 

Table 4 displays data on the changes in MetS over the two years. The odds of developing MetS increased in both the walnut and control groups (OR control = 1.11, 95% CI, 0.78-1.59, *P* = 0.56; walnut = 1.25, 95% CI, 0.89–1.75, *P* = 0.19) without significant between-group differences (*P* = 0.62). We found no difference in the results between men and women. Therefore, Table 4 presents the combined results.

Table 5 presents the composite scores for the incidence and reversion rates of MetS. There was a tendency towards decreased odds of reversing MetS in the walnut group as compared with the control group (OR walnut vs. control = 0.70, 95% CI, 0.31–1.58, *P* = 0.40). When compared to the control, the walnut group showed increased odds of developing MetS (OR walnut vs. control = 1.29, 95% CI 0.67–2.49, *P* = 0.44). Additionally, the two-year time period did not significantly alter the results for reversion and incident rates. It is worth mentioning that running the analysis without controlling for body weight did not change the results among all models. We have performed the analysis considering the location of the study. However, we found no significant difference in outcomes of metabolic syndrome and its individual outcomes between the USA and Spain cohorts. Therefore, data from both locations were combined to obtain more statistical power.

## 4. Discussion

The results of our study indicated that supplementing the diet of older adults with a daily dose of walnuts and no other dietary advice for two years resulted in neither an adverse effect on development of MetS nor its reversion. Even so, we observed a trend towards a beneficial effect of walnut consumption on blood pressure, a finding that has been reported in previous nut studies [11,31,32,33,34]. The results from the same WAHA feeding trial confirmed the favorable effects of walnut consumption in lowering blood pressure [31]. Other trials have also shown a decrease in blood pressure when consuming a Mediterranean diet (MedDiet) supplemented with 30 g/d of mixed nuts (walnuts 15g, hazelnuts 7.5g, and almonds 7.5g) [11,32]. The results from the 1999–2004 NHANES study of 13,292 participants indicated that nut consumption (≥ ¼ ounce/day) was associated with lower systolic blood pressure compared to non-consumers [33]. Another study found that participants who consumed walnuts (37 g/d) and a walnut oil diet (15 g/d) for six weeks experienced significant reduction in diastolic blood pressure [34]. This beneficial effect might be due to the PUFA, MUFA, and ALA content of nuts, which could synergistically interact to reduce the risk of high blood pressure [12]. Specifically, consuming a diet rich in ALA might increase arginine-vasopressin, thus lowering blood pressure [34]. Additionally, PUFAs and MUFAs can protect against high blood pressure by reducing the serum concentrations of the vasoconstrictor thromboxane-2 [12]. Moreover, walnuts are rich in potassium, which decreases the effect of angiotensin and relaxes smooth vascular muscle, resulting in enhanced peripheral vascular resistance and reduced risk of high blood pressure [35,36]. Nevertheless, a recent meta-analysis of randomized controlled trials with walnut-enriched diets failed to detect a blood pressure-lowering effect of walnut consumption [37].

We did not see a beneficial effect of walnut consumption on other components of MetS (triglycerides, BMI, HDL-C, blood glucose). Our findings are similar to those of a six-week randomized trial in Korea that found no significant difference in BMI, waist circumference, systolic blood pressure (systolic BP), HDL-C, triglycerides, and blood glucose between a control diet and a diet that was enriched with 30 g/day of mixed nuts (walnuts, peanuts, and pine nuts) [38]. The possible reasons for their null findings include small sample size and relatively short study duration. Additionally, a cross-sectional analysis of the Adventist Health Study-2 (AHS-2) showed that higher tree nut consumption (mean = 16 g/day) when compared to low (mean = 5 g/day) was not associated with significant differences in triglycerides, HDL-C, systolic and diastolic blood pressure, and blood glucose, albeit there was a trend towards lower MetS prevalence [39]. On the other hand, previous studies showed that consuming walnuts (1.5–2 ounces/day) could lower triglycerides among the hypercholesterolemic subjects [37,40,41,42,43]. A MedDiet that was supplemented with nuts showed a significant reduction in waist circumference [21,44] and serum triglycerides [44] by 5% and 10%, respectively. The results from a 24-week randomized trial in 60 free-living Asian Indians with MetS showed a significant improvement in waist circumference and blood glucose after the intervention [20% as pistachios from total energy intake (~49 g)] when compared to the control diet [45]. The mixed findings presented in the above studies could be due to multiple reasons, including nut dosage, duration of the study, and sample size, to mention a few. In the same cohort, we showed that SFA and MUFA decreased in both groups, but EPA + DHA intake were very similar between the walnut (mean = 0.23; 2 y change = 0.03) and control (mean = 0.23; 2 y change = 0.00) groups (*P* = 0.34) [46]. This could be a possible explanation why there was a decrease in HDL-C and triglycerides in both groups, but not significantly different between groups. Additionally, evidence showed that HDL-C gradually decreases with age [47].

Overall, our study showed that, when compared to a control diet, a walnut-enriched diet for two years resulted in no significant between-group differences in the development or reversion of MetS. Similar findings have been reported in other studies [21,39,48]. A multicenter randomized PREDIMED (PREvención con DIeta MEDiterránea) trial also found that the incidence rate of MetS did not differ between control participants and those assigned to the MedDiet plus nuts group (*P* = 0.3) [21]. A randomized clinical trial of 283 participants showed that a walnut diet for 12 weeks had no effect on incidence or reversion of MetS [19]. Another eight-week feeding trial with walnuts or cashews (corresponding to 20% of daily energy intake) showed no changes in MetS as compared to control (restricted from nut intake) [48]. In the much larger PREDIMED trial, however, the MedDiet with nuts was associated with MetS reversion [21]. Additionally, the SUN Project (Seguimiento Universidad de Navarra, University of Navarra Follow-up) assessed 9,887 participants for long-term nut consumption in relation to MetS and showed that participants who consumed ≥ 2 servings of nuts per week had 32% lower risk of incident MetS as compared to those who never/almost never consumed nuts. The inverse association between nut intake and MetS in this particular study was significant among women (*P* < 0.01) but not men [49]. The one-year PREDIMED results showed the prevalence of MetS was reduced by 13.7% in participants allocated to the MedDiet with nuts when compared to the control group (2.0%) [11]. 

Moreover, cross-sectional studies have shown that increasing nut consumption is associated with a decreased prevalence of MetS [33,50] and its components [33,39,50]. The results from the 1999-2004 NHANES study indicated that even small amounts of tree nuts (¼ ounce per day) could lower the prevalence of abdominal obesity, hypertension, hyperglycemia, low HDL-C and MetS [33]. In the Adventist Health Study-2, the prevalence of abdominal obesity was significantly lower among the participants who consumed more tree nuts (16 g/d) as compared to those with lower consumption (5 g/d) (*P* = 0.02) [39]. A cross-sectional assessment of the PREDIMED study also showed a 26% lower risk of harboring MetS in participants who consumed at least three servings of nuts per week when compared to those who consumed less than one serving per week [50]. 

Although there are no clear reasons for the null findings that were observed in our study, we speculate several factors that could explain the results. First and foremost, previous studies have shown that elderly individuals tend to lose lean body mass and gain fat over time [51,52,53], a phenomenon that might have occurred in this study with or without walnuts. Further, older subjects are more likely to develop cardiometabolic risk factors, making it challenging to observe the possible benefits of walnut consumption. Additionally, the original study excluded individuals with uncontrolled hypertension (on drug treatment for blood pressure ≥ 150/100 mmHg). It is possible that these individuals could have benefited from walnut consumption. Other reasons include the use of BMI as a surrogate for waist circumference and the lack of blinding to treatment. We did not adjust for caloric intake since this was a free-living study. In future walnut trials, researchers should consider providing nutritional advice for replacing calories from their normal intake with calories from walnuts. Additionally, results may be achieved by increasing dosage of walnuts and considering the possible dose response relationship between walnuts and MetS. Our study included measurements from baseline and at two years, but obtaining multiple measurements would best capture changes in MetS components. Notwithstanding these limitations, the present study was the largest and longest trial with walnuts and is the first to investigate the effect of walnut consumption on the development or reversion of MetS in a healthy elderly population.

## 5. Conclusions

Supplementing the diet of older adults with a daily dose of walnuts at 15% of energy resulted in no adverse effect on MetS or any of its components. A trend towards a beneficial effect in blood pressure was observed in the walnut group. Therefore, incorporating walnuts into the habitual diet of healthy elderly might lead to a favorable modification of risk factors that are associated with chronic diseases. Although the role of walnuts in modifying MetS components favorably needs further investigation, it is reasonable to recommend the inclusion of walnuts to the habitual diet of healthy elderly given that walnuts have been shown to lower the risk for CVD [43,44] and improve other health outcomes [11].

## Figures and Tables

**Figure 1 nutrients-12-00451-f001:**
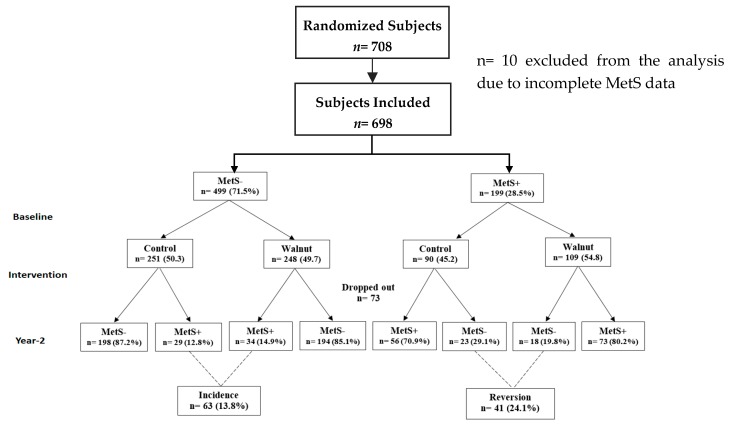
Classification of the development and regression rates of Metabolic Syndrome.

**Table 1 nutrients-12-00451-t001:** Baseline characteristics of participants by intervention groups.

Variables	Control		Walnut		Total (*n* = 625)
	MetS- (*n* = 199)	MetS+ (*n* = 107)	MetS- (*n* = 187)	MetS+ (*n* = 132)	
Center, *n* (%)					
LLU	101 (50.8)	44 (41.1)	97 (51.9)	57 (43.2)	299 (48.0)
BCN	98 (49.2)	63 (58.9)	90 (48.1)	75 (56.8)	326 (2.0)
Age, y, mean (SD)	68.7 (3.4)	69.1 (3.6)	69.4 (3.7)	69.2 (3.7)	69.1 (3.6)
Body weight, kg, mean (SD)	69.7 (13.0)	80.8 (15.7)	68.5 (12.3)	80.5 (16.7)	73.5 (15.2)
Gender, *n* (%)					
Women	145 (72.9)	64 (59.8)	125 (66.8)	86 (65.2)	420 (67.1)
Men	54 (27.1)	43 (40.2)	62 (33.2)	46 (34.8)	205 (32.9)
Race, *n* (%)					
LLU					
White	76 (75.3)	34 (77.3)	78 (80.4)	43 (75.4)	231 (77.3)
Hispanics	9 (8.9)	6 (13.6)	9 (9.3)	8 (14.0)	32 (10.7)
Black	6 (5.9)	3 (6.8)	7 (7.2)	5 (8.8)	21 (7.0)
Other	10 (9.9)	1 (2.3)	3 (3.1)	1 (1.8)	15 (5.0)
BCN (Caucasian)	98 (100)	63 (100)	90 (100)	75 (100)	326 (100)
Education, *n* (%)					
Secondary or less	83 (41.7)	45 (42.1)	71 (38.0)	55 (41.7)	254 (40.5)
Post-secondary	116 (58.3)	62 (57.9)	116 (62.0)	77 (58.3)	371 (59.5)
Ever smoker, *n* (%)					
Never	177 (88.9)	82 (76.6)	160 (85.6)	107 (81.1)	526 (84.2)
Yes	22 (11.1)	25 (23.4)	27 (14.4)	25 (18.9)	99 (15.8)
MetS Components, mean (SD)					
BMI, kg/m^2^	26.2 (4.2)	30.2 (4.0)	25.3 (3.3)	29.6 (4.4)	27.3 (4.5)
Systolic BP, mm Hg	125.2 (16.9)	130.9 (15.8)	122.8 (14.7)	131.6 (13.9)	126.8 (15.9)
Diastolic BP, mm Hg	77.0 (9.4)	80.0 (8.6)	75.6 (8.8)	81.3 (7.7)	78.0 (9.0)
Triglycerides, mg/dL	86.7 (31.5)	126.6 (51.5)	85.9 (30.1)	130.7 (57.5)	102.6 (46.5)
HDL cholesterol, mg/dL	60.8 (14.3)	51.0 (12.0)	59.3 (15.1)	49.8 (14.1)	56.4 (14.9)
Fasting serum glucose, mg/dL	93.5 (12.4)	105.6 (18.1)	92.5 (11.5)	107.1 (17.0)	98.1 (15.7)
Medications, *n* (%)					
Antihypertensive agents					
No	128 (64.3)	34 (31.8)	130 (69.5)	41 (31.1)	333 (53.3)
Yes	71 (35.7)	73 (68.2)	57 (30.5)	91 (68.9)	292 (46.7)
Antidiabetic agents					
No	196 (98.5)	94 (87.9)	183 (97.9)	114 (86.4)	587 (93.9)
Yes	3 (1.5)	13 (12.1)	4 (2.1)	18 (13.6)	38 (6.1)
Statins					
No	159 (79.9)	52 (48.6)	165 (88.2)	51 (38.6)	427 (68.3)
Yes	40 (20.1)	55 (51.4)	22 (11.8)	81 (61.4)	198 (31.7)
Other lipid lowering drugs					
No	197 (99.0)	99 (92.5)	186 (99.5)	126 (95.5)	608 (97.3)
Yes	2 (1.0)	8 (7.5)	1 (0.5)	6 (4.5)	17 (2.7)
Physical activity MET median (IQR)	2825 (2670)	2124 (2757)	2797 (2218)	2381 (2785)	2625 (2666)

Data are expressed as mean (SD), except for qualitative variables, which are expressed as *n* (%). MetS: metabolic syndrome. MET; physical activity metabolic equivalent, IQR; interquartile range.

**Table 2 nutrients-12-00451-t002:** Mean daily intake of selected nutrients in the walnut and control groups at two years.

Nutrients	Control (*n* = 312) Mean * (SD)	Walnut (*n* = 324) Mean * (SD)	*P*-Value **
Energy (kcal)	1632.5 (376)	1842.3 (442)	<0.0001
Total carbohydrate, g/d	186.8 (56)	189.6 (63)	0.621
Total protein, g/d	70.7 (18)	75.8 (18)	0.0003
Total Fat, g/d	66.8 (19)	89.3 (23)	<0.0001
SFA, g/d	19.2 (8)	20.8 (8)	0.005
MUFA, g/d	30.0 (11)	32.0 (12)	0.027
PUFA, g/d	11.5 (5)	29.6 (7)	<0.0001
α-Linolenic acid, mcg/d	1.1 (1)	4.6 (1)	<0.0001
Linoleic acid, mcg/d	9.7 (5)	24.3 (6)	<0.0001
Total fiber, g/d	19.1 (7)	22.0 (8)	<0.0001
Total carbohydrate (%E)	45.2 (8)	40.3 (7)	<0.0001
Total protein (%E)	17.6 (4)	16.6 (3)	0.001
Total fat (%E)	36.0 (6)	42.8 (6)	<0.0001
Total SFA (%E)	10.3 (3)	9.9 (2)	0.074
Total MUFA (%E)	16.3 (5)	15.3 (5)	0.013
Total PUFA (%E)	6.2 (2)	14.4 (3)	<0.0001

SFA, saturated fatty acids; MUFA, monounsaturated fatty acids; PUFA, polyunsaturated fatty acids; %E, percentage from total energy (kcal). * Mean of five unannounced 24-hour recalls or 3 food records at year 2. ** Two-sample *t*-test: *P* < 0.05, all variables were assessed for normality and transformations were utilized when required.

**Table 3 nutrients-12-00451-t003:** The effect of walnut consumption on the difference (year 2 minus baseline) of each component of metabolic syndrome (MetS).

Variables	Baseline	Year 2	Difference (year 2 Minus Baseline)		Group × Time Interaction *P*-value **
	Adjusted Mean (95% CI)	Adjusted Mean (95% CI)	Mean (95% CI)	*P*-value *	
**BMI**					
Control	27.5 (27.3, 27.7)	27.5 (27.3, 27.8)	0.03 (0.0, 0.06) ^a^	0.025	0.696
Walnut	27.1 (26.9, 27.3)	27.2 (26.9, 27.4)	0.04 (0.01, 0.07) ^a^	0.004	
**Systolic BP**					
Control	128.0 (126.4,129.7)	128.0 (126.3, 129.8)	0.01 (−1.7, 1.67) ^a^	0.99	0.265
Walnut	126.9 (125.3, 128.5)	125.6 (123.9, 127.3)	−1.30 (−2.9, 0.31) ^a^	0.114	
**Diastolic BP**					
Control	78.2 (77.3, 79.2)	78.2 (77.1, 79.2)	−0.08 (−1.07, 0.92) ^a^	0.88	0.369
Walnut	77.8 (76.9, 78.8)	77.1 (76.1, 78.1)	−0.71 (−1.68, 0.26) ^a^	0.15	
**HDL-C**					
Control	55.2 (53.8, 56.6)	52.4 (51.0, 53.8)	0.95 (0.93, 0.97) ^b^	<0.0001	0.836
Walnut	52.8 (51.6, 54.1)	50.0 (48.8, 51.3)	0.95 (0.93, 0.96) ^b^	<0.0001	
**Triglycerides**					
Control	93.2 (89.2, 97.3)	89.4 (85.3, 93.7)	0.96 (0.93, 0.99) ^b^	0.02	0.484
Walnut	96.6 (92.6, 100.7)	91.1 (87.1, 95.3)	0.94 (0.91, 0.98) ^b^	0.0007	
**FBG**					
Control	96.8 (95.5, 98.1)	95.7 (94.3, 97.2)	0.99 (0.98, 1.0) ^b^	0.086	0.194
Walnut	97.7 (96.4, 99.0)	97.8 (96.4, 99.2)	1.0 (0.99, 1.0) ^b^	0.923	

Mixed linear models adjusted for center, age, gender, weight, and log-physical activity; smoking status and antihypertensive drug use were included in the systolic and diastolic BP models; hypolipidemic drug use were included in the HDL and triglycerides models; and antidiabetic drug use were included in the FBG model. BP; blood pressure, HDL-C; high-density lipoprotein cholesterol, FBG; fasting blood glucose. * *P* value < 0.05 for comparing within subject different (year 2-baseline). ** *P* value < 0.05 for comparing between subject different (walnut vs. control). ^a^ Mean difference (95% CI) ^b^ Mean ratio (95% CI).

**Table 4 nutrients-12-00451-t004:** Marginal logistic model using generalized estimating equations (GEE) for MetS.

	Year 2 vs. Baseline OR (95% CI)	*P*-Value	Group × Time Interaction *P*-value
Control	1.11 (0.78, 1.59)	0.555	
Walnut	1.25 (0.89, 1.75)	0.192	0.62

Adjusted odds ratio for year 2 vs. baseline adjusted for center, age, gender, weight, and log-physical activity, smoking status; and antihypertensive, hypolipidemic, and antidiabetic drug use.

**Table 5 nutrients-12-00451-t005:** The odds ratio for reversion and development of MetS (walnut vs. control).

	Walnut vs. Control OR (95% CI)	*P*-value
Reversion rate	0.70 (0.31, 1.58)	0.395
Incidence rate	1.29 (0.67, 2.49)	0.441

Logistic regression adjusted for center, age gender, weight, log-physical activity, smoking status; and antihypertensive, hypolipidemic, and antidiabetic drug use.

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
