# Peer review of "Effects of Supplementing the Usual Diet with a Daily Dose of Walnuts for Two Years on Metabolic Syndrome and Its Components in an Elderly Cohort"

_nutrients, 2020, doi:10.3390/nu12020451_

Round 1

Reviewer 1 Report

I have revised the manuscript entitled “Effects of Supplementing the Usual Diet with a Daily 2 Dose of Walnuts for Two Years on Metabolic 3 Syndrome and its Components in an Elderly Cohort” I think that it is a very important intervention study on the effect of walnut supplementation on ordinary diets in old people. The manuscript includes a lot of variables that are very important to see the effect of including walnut in the diet. It also includes an important number of samples which is very important as other previous studies lack of samples. But as all these kinds of studies, the conclusions are quite complicated as a lot of factors can influence the results.  After the reading of the manuscript, some questions have arisen to me that I would like to comment with the authors and if possible I would like to know their opinion.

Line 181 “Even so, a significant decrease in plasma triglycerides and HDL-C was observed 180 in both groups (P < 0.05)”. This is said in the manuscript, but a not possible explanation is given. Why the same effect is observed in both groups? I see that in the control and walnut groups there are Mets+ and MetS-. I wonder if a comparison between the two groups has been done only with the Mets+ participants. I think that it would be interesting to see if the people with MetS+ along the study show different results at least in one of the parameters measured. Clarify, please. Another effect that has not been considered in the work is if the location has an effect on the study. We all know that the type of diets in the USA and Spain are different. Have you studied if there are differences in the results among the Barcelona and USA groups? Line 249 “Overall, our study showed that compared to a control diet, a walnut-enriched diet for 2 years 249 resulted in no significant between-group differences in the development or reversion of MetS. Similar 250 findings have been reported in other studies”. I wonder if a nutritional advised have been given to the walnut group if the results would have been the same. I see that total calorie intake in the walnut group was higher, It seems that they eat the same and include additionally the ingestion of walnuts. This effect may be masking the real effect of walnut consumption.

I think that it will be better if walnut is given instead of other foods keeping the total calorie ingestion at the same level. Do you think that the result would have been the same?

I am also a little bit surprised as the average energy intake obtained are lower than 2000 Kcal/day. It seems quite low, mainly in the USA as it is known that part of the population has a food intake higher than their needs.

Reviewer 2 Report

The manuscript entitled „Effects of Supplementing the Usual Diet with a Daily Dose of Walnuts for Two Years on Metabolic Syndrome and its Components in an Elderly Cohort” presents interesting data, however in my opinion manuscript cannot be published because the research design is not appropriate. The manuscript should be thoroughly revised, then Authors can resubmit the manuscript.

General

The study group should be homogeneous in terms of components of metabolic syndrome. In my opinion determination of MetS according to the Adult Treatment Panel III (ATP III) is inappropriate, because of the lack of the necessary criterion. As a result, confirming three out of five elements (various in case of various patients) causes that the study group very heterogenous. Moreover, according to these criteria, MetS may be diagnosed in patients with normal body weight. I suggest using the International Diabetes Federation criteria (2006) to determine MetS. Authors cannot “utilized BMI as a surrogate for waist circumference”, because they should rather determine obesity based on BMI - according to the IDF criteria (so it is the other issue causing that IDF criteria are for Authors more appropriate). Did the study participants follow a reduction diet? If they had excessive weight, they should use a reduction diet, so the results for the MetS group cannot be compared with control group as there were participants with a normal BMI. What diet did the study participants follow? Authors analyzed the consumption of walnuts, but what about the other sources of ALA in the diet, e.g. vegetable oils? What fats (oils) did they use? What was their consumption? Authors should present in details the intake of SFA, MUFA, PUFA, as well as the specific fatty acids, such as ALA. Authors wrote that: “Dietary intake was assessed using five unannounced 24-hour diet recalls (LLU) or 3-day food records obtained every 6 months (BCN) over the two years of the study.” Data from two different methods of data collection are combined into one group - this cannot be done. This is a methodological error. Data for the Loma Linda, California (LLU) and Barcelona, Spain (BCN) group should not be combined into one group, but they can be compared, but only if the same methods were used. Authors used inappropriate questionnaire to assess the level of physical activity. The Minnesota Leisure Time Physical Activity Questionnaire should be applied for adults up to 59 years of age, however, participants over 60 years of age participated in this study. Authors should use the questionnaire to assesses frequency and duration of various physical activities typically undertaken by older adults for example Physical Activity Scale for the Elderly (PASE) or Yale Physical Activity Survey (YPAS). In addition, authors must clarify in which season the data were collected. In conclusion, Authors should focus on analyzing data only for patients with metabolic syndrome to determine time differences, as well as to present separately the results for the American and Spanish group.

Other comments:

Results are not clearly presented. In Table 1 Authors presented 4 groups (control with MetS, control without Met, walnuts with MetS and walnuts without MetS), but afterwards in Table 2 Authors presented 2 groups only (control and walnuts). Authors should present the number of ethical committee's approval. How did Authors verify the normality of distribution of data? What tests did they use? In section Statistical Methods and Analyses Authors should complete the information about the statistical tests that have been applied for example Two-sample t-test. Under tables 3-5 there is no information about the statistical tests. Authors should follow the instructions for authors (the way of presenting references is incorrect). Author Contributions is incorrectly presented. Authors should follow the instructions for authors.

Reviewer 3 Report

The manuscript “Effects of Supplementing the Usual Diet with a Daily 2 Dose of Walnuts for Two Years on Metabolic....” reports the design and outcomes of a 2-year, dual center, randomized clinical trial, where a total number of 698 healthy elderly participants were randomly assigned to either a walnut supplemented or a control diet. Outcomes were that the supplementation of the diet with a daily dose of walnuts had no effect on MetS status or any of its components. The walnut group however tended to have lower blood pressure.

The paper is well-written with a suitable Introduction and a good study design. However, some parts require amendments.

2. Materials and Methods

L.74: you first mention a total number of 708 elderly people, then in Fig. 1 a total of 698 subjects is shown. Please, check this datum or make clearer.

3. Results

This section needs revision, as in general tables and figures are poorly presented. In particular, I suggest adding some considerations about results presented in Figure 1. As mentioned above, please, also check or clarify the total of subjects  (i.e., n=698).

Table 1: amend layout, please, so that the different sections (i.e., Gender, Race, the two centers, etc.) are more evident at a glance.

L.185. please move up in line 184.

Table 3: make clearer the differences between men and women.

5. Conclusions

L. 293 You mention “no adverse effect on MetS”. The study better highlights no effect on MetS status or any of its components.

I suggest adding the conclusions included in the abstract (ll. 30-33) which are more faithful to the outcomes of the study.

Why are you suggesting walnut intake should be recommended, when the main outcomes of your study on the health effect of walnut intake is so weak? Please, delete or rephrase the last sentence (ll. 295-296).

Round 2

Reviewer 2 Report

In my opinion, patients should be divided into two groups (1) with normal body weight and (2) with excessive body weight. Including patients with normal and excessive body weight in one group is not correct. I would like to know why patients with BMI over 40 were excluded from the study? Obesity is one of the parameters of the metabolic syndrome.

Line 156-161 There is incorrect text font.

References and Author Contributions must be corrected.

Reviewer 3 Report

Dear Authors, thank you for amending the manuscript according to suggestions.

Author Response

Your comments and suggestions help us to improve the paper.

Thank you very much.